# Is half the world's population really below 'replacement-rate'?

Stuart Gietel-Basten[1]◉*, Sergei Scherbov[2]◉

**1** Division of Social Science, The Hong Kong University of Science and Technology, Clear Water Bay, Kowloon, Hong Kong SAR, P.R. China, **2** World Population Program, International Institute for Applied Systems Analysis, Laxenburg, Austria

◉ These authors contributed equally to this work.
* sgb@ust.hk

## Abstract

A perennial activity of demographers is to estimate the percentage of the world's population which is above or below the 'replacement rate of fertility' [RRF]. However, most attempts to do so have been based upon, at best, oversimplified, or at worst, simply incorrect assumptions about what RRF actually is. The objective of this paper is to calculate the proportion of the world's population living in countries with observed period total fertility rates [TFR] below the respective calculated RRF, rather than the commonly used measure of 2.1. While the differences between comparing TFR to 2.1 or RRF are relatively modest in many periods when considering populations at the national level, a significant difference can be observed in the near future based upon India's fertility and mortality trajectories. Our exercise represents a means of 'correcting the record' using the most up-to-date evidence and using the correct protocol.

**Data Availability Statement:** The data derived from the United Nations World Population Prospects (2017 Revision) used in this paper (e.g. TFR, life tables) are publicly available as a downloadable R package at https://cran.r-project.org/web/packages/wpp2017/index.html. The Indian

## Introduction

A perennial activity of demographers is to estimate the percentage of the world's population which is above or below the 'replacement rate of fertility' [RRF]. One of the first major pieces to do this was Chris Wilson's 2004 'Letter' in *Science* [1]; with a more elaborated version written by Wilson with Gilles Pison published the following year [2]. The main conclusion of Wilson's study [1] was that:

> In assessing the state of the planet, it is important to note that during late 2003 or early 2004, the human population will cross a historic, but so far largely unnoticed, threshold. Most of the world's population either already do, or soon will, live in countries or regions in which fertility is below the level of long-run replacement [1].

This conclusion was widely reported and cited. To take a few illustrative examples, Myrskylä et al. [3] cite the study in their influential 2009 *Nature* paper on the relationship between fertility and Human Development Index, stating that 'more than half of the global population now lives in regions with below-replacement fertility (less than 2.1 children per woman).'

life tables used in this paper are available from the Office of the Registrar General and Census Commissioner, India, at http://www.censusindia. gov.in/Vital_Statistics/SRS_Life_Table/SRS% 20based%20Abridged%20Life%20Tables% 202013-17.pdf.

**Funding:** The author(s) received no specific funding for this work.

**Competing interests:** The authors have declared that no competing interests exist.

Again, citing Wilson [1], Keyfitz and Caswell's 2005 textbook [4] stated that 'it is now estimated that, as of early 2004, more than half of the world population now lives in countries or regions where fertility is below replacement level.'

In his discussion of the historical antecedents of below-replacement fertility, Wilson [1] states that 'In the early 1950s, below-replacement fertility was virtually unknown. By the late 1970s, there had been considerable change, with about a quarter of the world's population experiencing fertility below 2.1.' This figure of 2.1 is, 'conventionally regarded as replacement level in conditions of low mortality (that is, where life expectancy is 70 years or more)' [1]. While the elaborated version of the study [2] does admit that 'strictly speaking, a precise measure of replacement would use the exact mortality level and sex ratio at birth for each country to calculate replacement level' 2.1 is still used as 'a convenient overall estimate' (p.2).

A similar exercise is found in the *Key Findings and Advance Tables* report of the 2017 *Revision* of the *World Population Prospects*. In one figure in particular, the distribution of the world's population by level of total fertility classified as: 'high fertility', 'intermediate fertility' and 'below replacement fertility (less than 2.1 births per woman)' [5] for various time points in the past and future. This figure was reproduced in a *Population Facts* media/policy brief [6]. Again, this report and the conclusions was cited and disseminated widely. In a 2017 piece entitled 'Half the world's population reaching below replacement fertility' published in *nIUSSP*, Tomas Frejka [7] wrote that 'According to the most recent UN estimates . . .almost one half of the world's population lives in countries with below replacement fertility . . . i.e. with a [TFR] below 2.1 births per woman'. The figures were quoted in a 2018 think piece published in *Yale Global Online* by Chamie [8] as well as many media articles.

The final example of the proportion of the world's population 'living below RRF' can be found in the Global Burden of Disease Study new estimate of TFR for all countries [9]. Commenting on the findings, the lead author stated that 'We've reached this watershed where half of countries have fertility rates below the replacement level, so if nothing happens the populations will decline in those countries . . .It's a remarkable transition . . .It's a surprise even to people like myself, the idea that it's half the countries in the world will be a huge surprise to people" [10]. In this study, RRF is determined as 'the TFR at which a population replaces itself from generation to generation, assuming no migration; generally estimated to be 2.05' [9].

While not an exhaustive list by any means, these three examples show how this exercise has been performed by leading population scholars; published in *Science*, the *Lancet* and UN Reports; and disseminated very widely within academic, policy and popular discourse.

The problem is that these comparisons are, strictly speaking, not entirely accurate.

Leaving aside for a moment the benefits of comparing to cohort rather than period TFR to RRF [11], each of these estimates assumes (or at the very least *infers)* a constant RRF over time and space to compare against. It is simply not the case that RRF is 2.1 (or 2.05) in all countries of the world today; and it is certainly not the case for the past [12]. RRF is determined by mortality among women of childbearing age and the sex ratio at birth. In a world where mortality during child rearing ages coexists with skewed sex ratios at birth in some countries, contemporary RRF in all countries simply cannot be 2.05, or even 2.1 for that matter. For the period 1995–2000, Espenhade and colleagues [12] calculated that *RRF* at the global level was 2.34; ranging from 2.09 in 'More Developed Regions' to 2.70 in Africa, with Sierra Leone having the highest RRF in the world at 3.43. Looking into the past, RRF in England and Wales in 1910 was around 2.6 [11].

In this paper, then, we try to *better* estimate the proportion of the world's population which lived in settings with a period TFR of either below or above the same country's *actual* RRF (rather than a TFR of 2.1), and explore how these change over time. Referring back to the estimates made by Espenhade et al. [12] above, then, the data point for Sierra Leone in 1990–95

will be a comparison of *its* TFR for that time period and not 2.1, but rather *its'* calculated RRF for that time, namely 3.43.

## Materials and methods

We produce estimates of RRF on a quinquennial basis for the period 1950–2100. Utilizing data on sex ratios at birth, and both past and projected life tables, we are able to compute *RRF* for all currently defined territories of the world according to the following formula. The underlying principle of finding the TFR at which replacement level fertility occurs is, simply, 1 divided by the Net Reproduction Rate [NRR]. This is calculated as follows:

$$NRR = Sr\sum_{x=15}^{50} L(x) * F(x) \tag{1}$$

where *Sr* is the proportion of girls born, *L(x)* is person years and *F(x)* = age specific fertility rates. *F(x)* = *f(x)*\**TFR*, where $\sum_{x=15}^{50} f(x) = 1$. Thus, the TFR that produces replacement level of fertility is equal to:

$$RRF = 1/(Sr\sum_{x=15}^{50} L(x) * F(x)) \tag{2}$$

The first illustration will be a comparison of TFR (derived from [13]) to the calculated RRF of all (current) UN defined territories, resulting in a plot of the cumulative population of territories ordered by distance away from RRF. Rather than a fixed line at ~2.1, this will more accurately represent the number of people living in territories with TFR above or below RRF.

The second illustration is based upon the measurement protocol implemented in Wilson [1]. By disaggregating China and India into their constituent states/provinces, and distributing them according to the TFR as above or below a defined RRF of 2.1, Wilson was able to claim that a majority of the world's population made this transition to below RRF in the early 2000s. When measured by country this transition has not yet occurred. The TFR in almost all provinces of China today have made the transition to below a threshold 2.1 [14]. The states of India, with its 2017 population of 1.339 billion, however, do straddle the 2.1 threshold. As James [15] observes 'Of the 20 states constituting nearly 98% of the population in the country, 11 states with 46% of the total population achieved a NRR of 1.0, implying replacement-level fertility'. As such, in the spirit of Wilson [1], we attempt to disaggregate the states of India, and count them as 'countries' and compare them to 2.1 as well as their own calculated RRF. (Of course, other large countries, such as Indonesia, which straddle a fertility of around 2.1 could be disaggregated; but this is beyond our scope here).

Our primary data source in this paper is the *2017 Revision* of the United Nations' *World Population Prospects* [13]. Although there are clearly challenges and issues with the data, it is still the predominant and most widely used set of demographic indicators currently in use. For the second illustration, we will use ASFRs and TFRs for the Indian States derived from the 2016 Sample Registration Survey (see Table 1 below) [16]. We will calculate RRFs for each state according to the protocol above using life tables derived for the period 2012–16 [17]. Total population figures are derived from Indian Government projections for the year 2017 [18].

Because of the lack of available data (namely state-level life tables for the historical period), however, we are only able to perform this exercise for one, contemporary time period. Clearly, these different sources and dates are a major limiting factor in this illustration. A further issue is that these subnational TFR and life table data are provided only for the 'larger states' of India. This means that we are estimating the RRF (and comparing to the total population) of the sum of these larger states, rather than of India as a whole. Despite this, these 'larger states'

**Table 1. Indian States, observed TFR (2016), RRF (2012–2016), and difference between the two, projected population (2017), ordered by observed TFR.** Sources: Authors calculations based on [16–18].

| | Estimated Population | Cumulative sum of states where observed TFR <2.1 | (a) Observed TFR | (b) Calculated RRF | TFR-RRF |
|---|---|---|---|---|---|
| Delhi | 18,110,349 | 18,110,349 | 1.6 | 2.26 | -0.66 |
| West Bengal | 96,775,592 | 114,885,941 | 1.6 | 2.17 | -0.57 |
| Tamil Nadu | 75,844,451 | 190,730,392 | 1.62 | 2.18 | -0.56 |
| Andhra Pradesh | 52,375,124 | 243,105,516 | 1.67 | 2.21 | -0.55 |
| Punjab | 29,344,896 | 272,450,412 | 1.67 | 2.24 | -0.57 |
| Himachal Pradesh | 7,246,418 | 279,696,830 | 1.68 | 2.21 | -0.54 |
| Jammu & Kashmir | 13,477,325 | 293,174,155 | 1.69 | 2.24 | -0.55 |
| Kerala | 35,043,531 | 328,217,686 | 1.76 | 2.09 | -0.33 |
| Maharashtra | 119,581,739 | 447,799,425 | 1.78 | 2.23 | -0.45 |
| Karnataka | 65,426,566 | 513,225,991 | 1.8 | 2.19 | -0.39 |
| Uttarakhand | 29,344,896 | 542,570,887 | 1.85 | 2.29 | -0.44 |
| Odisha | 44,912,901 | 587,483,788 | 1.96 | 2.24 | -0.29 |
| Gujarat | 63,000,000 | 650,483,788 | 2.24 | 2.34 | -0.1 |
| Haryana | 27,443,256 | 677,927,044 | 2.28 | 2.37 | -0.09 |
| Assam | 34,068,394 | 711,995,438 | 2.32 | 2.33 | -0.01 |
| Chhattisgarh | 28,125,421 | 740,120,859 | 2.48 | 2.22 | 0.26 |
| Jharkhand | 36,672,687 | 776,793,546 | 2.63 | 2.26 | 0.38 |
| Rajasthan | 76,802,294 | 853,595,840 | 2.69 | 2.37 | 0.32 |
| Madhya Pradesh | 80,894,777 | 934,490,617 | 2.81 | 2.32 | 0.49 |
| Uttar Pradesh | 224,558,257 | 1,159,048,874 | 3.11 | 2.39 | 0.72 |
| Bihar | 117,153,097 | 1,276,201,971 | 3.32 | 2.26 | 1.06 |
| India | 1,276,201,971* | | 2.26 | 2.28 | -0.02 |

*This figure is the sum of these 'larger states'.

See Methods section for elaboration.

account for over 96% of the total population of India. We proceed, then, with these caveats about the veracity of this second illustration.

## Results

Our overall set of illustrations identify a difference between RRF and the conventional measure of a TFR of 2.1 over time and space. In 1950–55, for example, global RRF was, in fact, 2.96, rather than 2.1. In Sub-Saharan Africa during the same period, it was as high as 3.64; while in Northern Europe it was just 2.16. For the contemporary period (2010–2015), the global RRF was 2.29, with a range from 2.06 in Macao SAR and Luxembourg up to around 2.7 in Chad, Central African Republic and Nigeria.

The results of the first illustration are shown in Fig 1 and Fig 2 for selected years. Fig 1 represents the total population of the world (y-axis), while the x-axis is ordered by the observed TFR in each territory. The line intersecting the x-axis represents the 'conventional' measure of RRF, namely 2.1 –akin to the graphs in the studies discussed in the Introduction. The line intersecting with the y-axis, meanwhile, represents the point at which the cumulative population plot reaches 2.1. The number on that line represents the proportion of the global population living in territories where the observed TFR is at, or below, the conventional measure of 2.1. This figure shows that during the period 2015–2020 the percentage of the world's population living in countries with an observed TFR of 2.1 or below is 49%. By the period 2020–25,

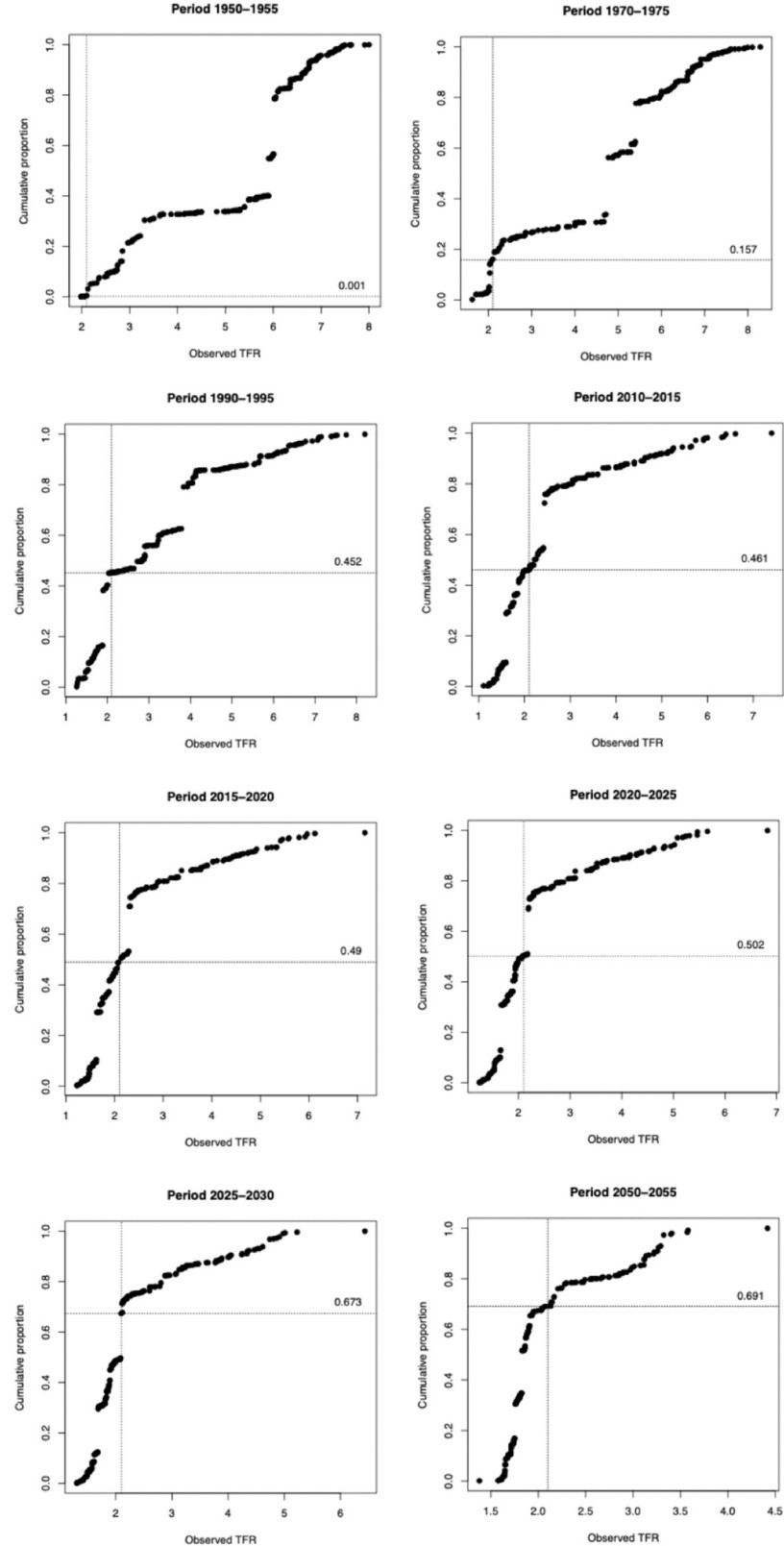

**Fig 1.** Cumulative proportion of global population (y-axis) of UN-specified territories ordered by observed TFR (x-axis) with proportion living in territories where observed TFR is equal or less than 2.1 (intersecting line), selected years.

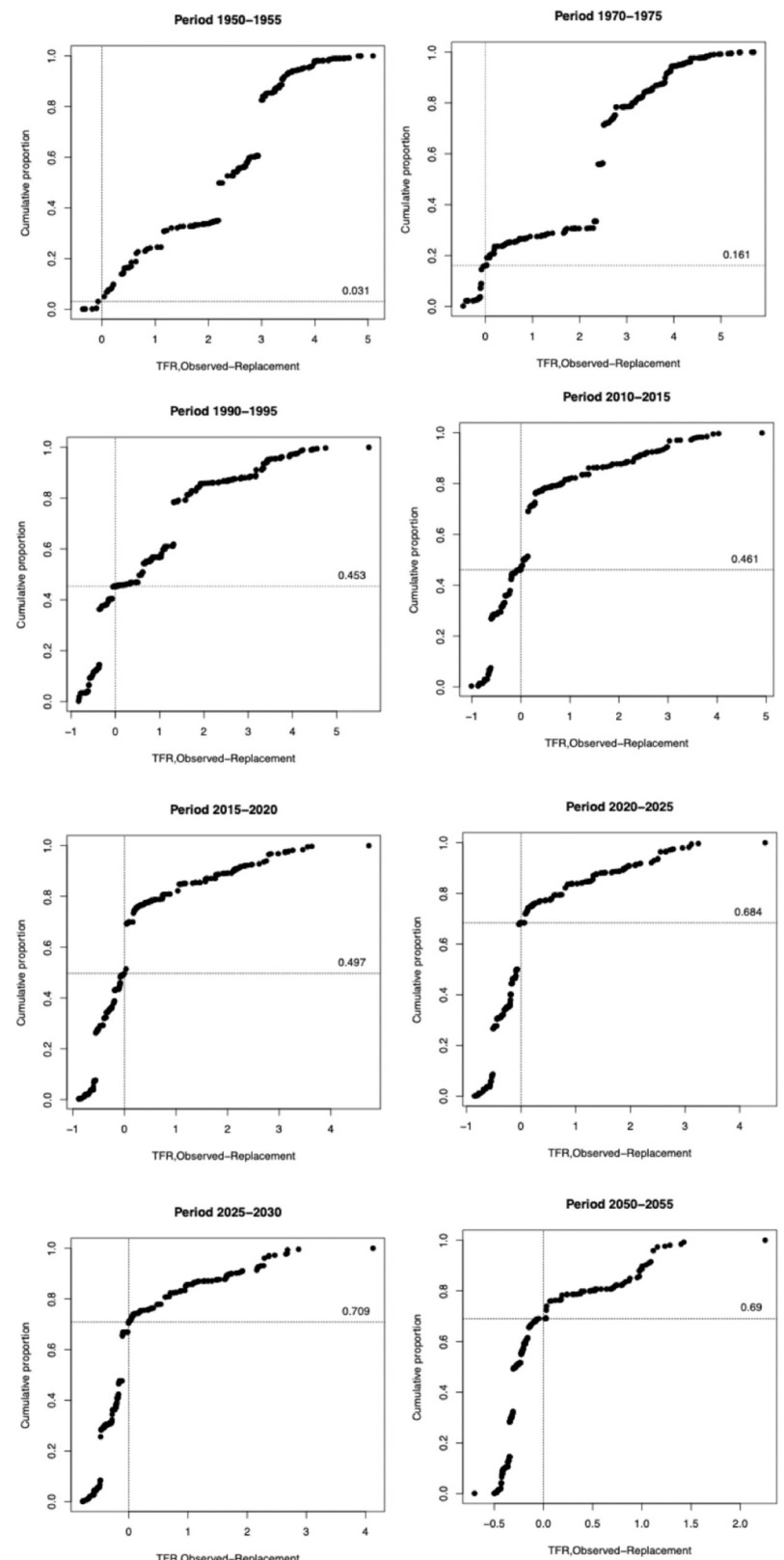

**Fig 2.** Cumulative proportion of global population (y-axis) of UN-specified territories ordered by observed TFR *minus* calculated RRF (x-axis) with proportion living in territories where observed TFR is equal or less than the point at which observed TFR = calculated RRF (intersecting line), selected years.

this will have increased to just over 50%. By 2025–30, the percentage increases sharply to 67.3% as a result of the projected TFR decline of India. Over the following decades the rate of increase will be much slower, reaching 69.1% by 2050–55.

Fig 2 also has the cumulative proportion of global population on the y-axis, but this time the countries are ordered by their observed TFR *minus* the calculated RRF for that country. The line intersecting the y-axis at zero, therefore, represents the point at which observed TFR is equal to the calculated RRF. Again, the y-intercept represents the cumulative population below this line. This is because of a number of relatively countries whose observed TFR was slightly higher than 2.1, but lower than their own calculated RRF.

Comparing Fig 1 and Fig 2 we can note some key differences and similarities. In 1950–55, the cumulative proportions in Fig 2 are significantly higher than in Fig 1 (even if still small in an absolute and relative sense compared to later years. For the periods leading up to 2010–15, the differences between a base comparison between a TFR of 2.1 and the calculated RRF are small, accounting for just a few percentage points–even being statistically the same in 2010–2015. In the most recent time period, using the calculated RRF leads to a slightly greater cumulative proportion living in countries with 'below replacement fertility'; but still just short of a majority. For the period 2020–25, however, there is a very sizeable difference–a cumulative proportion of 50.2% in Fig 1 compared to 68.4% in Fig 2. This is primarily because while the UN forecasts Indian observed TFR to drop below 2.1 during the period 2025–30, it is forecast to drop below its *calculated RRF* of 2.23 during this earlier time period. By 2025–30 the difference declines again, back to 3.6% and is almost identical in 2050–55. This illustration emphasizes the critical role that India, one of the world's demographic billionaires, will play in determining the point of crossover to a majority of the world population living in territories with 'below replacement fertility'.

Table 1 shows the major states of India by TFR. Immediately we can see that a number of states (including Delhi, Karnataka, Kerala, Tamil Nadhu and West Bengal) have observed TFRs below the conventional measure of 2.1. If we take the cumulative population of these states (essentially up to and including Odisha) using the data in column (a), the cumulative population is 587.48 million–or 46.0% of the total population under analysis. If, however, we turn to section (b) of the table, we can immediately see that actual RRF for all Indian states is not 2.1. In particular, we can see that there are certain states where the observed TFR is *greater* than the conventional measure of 2.1, but *lower* than the actual calculated RRF. These states are Assam, Gujarat, and Haryana; with 2010–15 populations of 34.07 million, 63.00 million and 27.44 million respectively. By including these three states in the 'below replacement fertility' category, this takes the cumulative population living in such states up to 712.00 million–or 55.8% of the population under review here.

## Discussion

The question at the heart of our paper is 'is the majority of the world's population living in countries with below replacement fertility; and if not, when. The simple answer is: it depends.

If we consider country-level units alone, the simple answer is 'no' for the present whether we base our calculation on the conventional measure of 2.1 or on the actual calculated RRF. However, because Indian observed TFR is projected to fall below its actual RRF, but not below 2.1 during the period 2020–25, there is a natural discrepancy between the answer which these two approaches would give. If we use the conventional TFR, then the period where the cumulative proportion is greater than half will be in 2025–30. If we use the actual calculated RRF it will be 2020–25.

The evidence from the second illustration shows us clearly just how India is straddling the boundary of replacement fertility–*however defined*. By adopting the conventional measure of

2.1, then around 46% of the population of India is living in 'below replacement states.' When we compare to the calculated actual RRFs, meanwhile, we can say that just over 55% live in 'below replacement states' (because of the three states where observed TFR is greater than 2.1 but lower than replacement).

Fig 1 indicates that, in the period 2015–20, 49.0% and 49.7% of the world's population lived in territories characterised with observed TFRs below either 2.1 or the actual calculated RRF respectively. Numerically, this translates into 3.72 and 3.77 billion respectively (based on a global population of 7.59 billion [13]). If we add on the figures generated for the Indian states in the second illustration (i.e. 587.48 million and 712.00 million) we arrive at 4.32 billion people living in territories or Indian States with observed TFR below 2.1 and 4.48 billion people living in territories or Indian States with observed TFR below their actual calculated RRF. Going back into percentage terms, these translate to 56.74% and 59.08% respectively. Therefore, when we subdivide India, the majority of the world does, indeed, live in territories with below-replacement fertility *however defined*.

Our exercise, of course, has a number of limitations. As observed earlier, the data limitations of any reconstructions of past populations are well known; as, indeed, are the vagaries of trusting in population projections in an unquestioning manner. Despite this, we still note that the datasets we use–especially those from the United Nations–are the best extant corpus of evidence upon which to base our analysis.

A second limitation is the arbitrary choice of splitting up India into its constituent parts. Of course, other countries have regions whose observed TFRs straddle either 2.1 or, inevitably, their calculated RRF. Indonesia is such an example of a very large country characterised by such regional heterogeneity. However, the demographic predominance of India in terms of its sheer size does, we suggest, justify the focus on it in this second illustration.

A final issue relates to the fact that TFR is distorted by changes in timing of births, duration of marriage, parity distribution and at entry into sexual union–known as the 'tempo effect'. Naturally, RRF is affected by these factors because the RRF is estimated from the period ASFR and TFR of a given year. In order to take account for this it is preferable to use the tempo-adjusted TFR for computing the RRF which may well have an effect on the proportion of world population below the replacement level (or, effectively, the adjusted RRF). However, tempo-adjusted fertility rates are not available for all countries of the world (or, indeed, the states of India) at present, either for estimates of the past or projections for the future.

Finally, some readers are likely to view this our exercise here is both pedantic and pointless. It could be seen as pedantic in that it is, in some ways, arguing the toss between a 'general' and a 'precise' definition of RRF (or even an absolute versus a relative measure) which, at least in contemporary societies, is only a narrow numerical difference anyway. It could also be seen as pointless inasmuch as the size of the global population (divided on broadly Westphalian notions of citizenry) living either above or below replacement level is a fairly meaningless, arbitrary distinction which has little or no meaningful consequence. In fact, our findings do not materially differ from those examples presented in the Introduction, so again–what is the point?

As far as the latter critique is concerned, we would probably be inclined to agree. Many people obsess over the replacement rate, often presenting it as some kind of 'optimal.' The conditions of optimality are, however, so narrowly defined as to render it of only limited practical use [19]; especially when taking into account migration and human capital. Indeed, the language around the replacement rate as some kind of 'target' is arguably deleterious to the construction of multidimensional, holistic population policies which genuinely address contemporary social, economic and political challenges [20–21].

However, as for being pedantic, we might be a little more cautious. Demography is, at its heart, a mathematical discipline. Rather than being grounded in a kind of 'statistical relativism', we should surely aspire to be disciplined in our what we calculate, and how we proceed to interpret, frame and present these calculations. The presentations of the 'percentage of the world's population living under replacement rate fertility' given in the three examples set out in the Introduction can either be called factually incorrect, or sloppily conceived. This should be corrected. While our own results can certainly be questioned in terms of the various limitations we outline above, we can at least be confident that we have followed a correct demographic protocol to arrive at these figures. Simply as a point of 'correcting the record' of a widely disseminated set of figures which have been published in the very highest ranked journals, we argue that this exercise can be justified. While most studies call for further research in a given area, we simply argue the research that is done is done correctly, and that any shortcuts we make should be very clearly defined and better justified.

## Author Contributions

**Conceptualization:** Stuart Gietel-Basten, Sergei Scherbov.

**Data curation:** Stuart Gietel-Basten, Sergei Scherbov.

**Formal analysis:** Stuart Gietel-Basten, Sergei Scherbov.

**Investigation:** Stuart Gietel-Basten, Sergei Scherbov.

**Methodology:** Stuart Gietel-Basten, Sergei Scherbov.

**Project administration:** Stuart Gietel-Basten, Sergei Scherbov.

**Resources:** Stuart Gietel-Basten, Sergei Scherbov.

**Validation:** Stuart Gietel-Basten, Sergei Scherbov.

**Visualization:** Stuart Gietel-Basten, Sergei Scherbov.

**Writing – original draft:** Stuart Gietel-Basten, Sergei Scherbov.

**Writing – review & editing:** Stuart Gietel-Basten, Sergei Scherbov.

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
