## [Decision Letter · Decision Letter 0]

21 Aug 2019

PONE-D-19-14424

Is half the world’s population really below ‘replacement-rate’?

PLOS ONE

Dear Authors,

Thank you for submitting your manuscript to PLOS ONE. After careful consideration, we feel that it has merit but does not fully meet PLOS ONE’s publication criteria as it currently stands. Therefore, we invite you to submit a revised version of the manuscript that addresses the points raised during the review process.

We would appreciate receiving your revised manuscript by 21 October 2019. To enhance the reproducibility of your results, we recommend that if applicable you deposit your laboratory protocols in protocols.io, where a protocol can be assigned its own identifier (DOI) such that it can be cited independently in the future. For instructions see: http://journals.plos.org/plosone/s/submission-guidelines#loc-laboratory-protocols

We look forward to receiving your revised manuscript.

Kind regards,

Professor Hafiz T.A. Khan, Ph.D

Academic Editor

PLOS ONE

Journal Requirements:

2. Please ensure that you refer to Figure 2 in your text as, if accepted, production will need this reference to link the reader to the figure.

3. Please upload a copy of Figure 4, to which you refer in your text. If the figure is no longer to be included as part of the submission please remove all reference to it within the text.

Additional Editor Comments:

Minor revision.

Reviewers' comments:

Reviewer's Responses to Questions

**Comments to the Author**

1. Is the manuscript technically sound, and do the data support the conclusions?

Reviewer #1: Yes

Reviewer #2: Yes

2. Has the statistical analysis been performed appropriately and rigorously? 

Reviewer #1: Yes

Reviewer #2: Yes

3. Have the authors made all data underlying the findings in their manuscript fully available?

Reviewer #1: No

Reviewer #2: No

4. Is the manuscript presented in an intelligible fashion and written in standard English?

Reviewer #1: Yes

Reviewer #2: Yes

5. Review Comments to the Author

Reviewer #1: This paper deals with an interesting topic, i.e. evaluating the variability in contemporary replacement level of fertility instead of empirical estimated level. The paper justified the findings through illustration from different regions. The only flaw of the article is that it is too brief in some places. The following comments are my recommendations about some specific points.

1. The paper requires a careful revision for small typos.

2. TFR=2.1 is usually considered as replacement level of fertility in conventional setting (also in this paper). This value should be denoted using a consistent term all over the paper (say BRF, or some new term).

3. Please summarize the contents and explanations of the graph of Wilson in one or two sentences.

4. In addition to the three sketches, please include few more lines mentioning the importance of a variable replacement level of fertility (say, for national level policy implication).

5. In materials and methods, interpretation of NRR, f(x) and L(x) are missing.

6. Please express all the demographic indicators like TFR, RRF,... in plain texts (instead of italic) all over the paper.

7. There should be two distinct part in the Materials and method section. The first one will be for methods only and the second one will be for data sources with justification notes (where necessary). The word "exercise" should be replaced by illustration or analysis.

8. The statement "This means that 54,364,244 are, effectively, unaccounted for." is not clear.

9. In the result section, the observed TFR should be denoted with a distinct, different notation.

10. The result analysis should be stronger and it could benefit from few more graphical representation (please see the next comments). For instance, the difference between the observed TFR and RRF could be presented from other different point of views. Figure 1 is difficult to understand or compare. A plot with higher resolution is required for that.

11. The statement "Taken together, it is reasonable to assume that the majority of the world's population ... probably also TFR<2.1." is not clear. Please rephrase with further explanation. Beside, this statement is too generalizing without mentioning clear assumption or citing reference.

12. The first part of the result needs more explanation for the observed regional pattern. This global illustration is more convincing for the title of the paper than Indian or Chinese context alone. Both for the global and Indian data, a plot showing the seasonal difference (for selected years) between contemporary replacement level of fertility and RRF will be useful.

13. From the formula of RRF, clearly the variability in replacement level of fertility are attributable to its components (SR_b, L(x) or ASFR). How the different components are affecting the regional scenario? I think including these component-wise detailed findings is out of scope of the current manuscript but some discussion on this should be included.

14. The discussion section will be better with some guidelines regarding further scope of research from the obtained results.

Reviewer #2: The is an interesting paper. I agree with the arguments made by the author. However, I have some minor comments which the author may consider it before the publication.

1. I am unable to get the answer in the conclusion about the title you started with. Is half world population really below replacement level? The analysis and the conclusion seems to be brief and unable to get the correct view of the authors based on the revised calculations. Though the authors have contested the other publications and methods those authors have used for their conclusions in the introduction, I am unable to get answer for it. The figure 1 is not clear to read, proportion of world population below the replacement level over time based on the revised method. It would be good, if you provide a table comparing your proportion of pop with previous authors calculations to prove that how your method has changed the previous authors conclusions in a comparative view. In the case of India, it is clearly shows that proportion below RRF is greater than proportion of pop below the TFR of 2.1. But this was not the case with world population. It is very unclear.

2. Though I agree with the revised method of RRF to be used for the replacement level fertility not TFR 2.1. But I have a concern, how RRF is affected by the tempo effect for a given period. We know that TFR is distorted by changes in timing of births, duration of marriage, parity distribution and at entry into sexual union. How the RRF is affected by these factors because the RRF is estimated from the period ASFR and TFR of a given year. Suppose we use the tempo adjusted TFR for computing the RRF, what would be effect on the proportion of world population below the replacement level (adjusted RRF). In other words, how it would affect your conclusions? What would be your response on this?

6. PLOS authors have the option to publish the peer review history of their article (what does this mean?). If published, this will include your full peer review and any attached files.

Reviewer #1: Yes: Dr. Ahbab Mohammad Fazle Rabbi

Reviewer #2: Yes: Kannan Navaneetham

---

## [Author Response · Author response to Decision Letter 0]

15 Oct 2019

Please see attached response to reviewers file.

---

## [Decision Letter · Decision Letter 1]

28 Oct 2019

Is half the world’s population really below ‘replacement-rate’?

PONE-D-19-14424R1

Dear Authors, 

We are pleased to inform you that your manuscript has been judged scientifically suitable for publication and will be formally accepted for publication once it complies with all outstanding technical requirements.

With kind regards,

Professor Hafiz T.A. Khan, Ph.D

Academic Editor

PLOS ONE

Additional Editor Comments (optional):

Reviewers' comments:

Reviewer's Responses to Questions

**Comments to the Author**

1. If the authors have adequately addressed your comments raised in a previous round of review and you feel that this manuscript is now acceptable for publication, you may indicate that here to bypass the “Comments to the Author” section, enter your conflict of interest statement in the “Confidential to Editor” section, and submit your "Accept" recommendation.

Reviewer #1: All comments have been addressed

Reviewer #2: All comments have been addressed

2. Is the manuscript technically sound, and do the data support the conclusions?

Reviewer #1: Yes

Reviewer #2: Yes

3. Has the statistical analysis been performed appropriately and rigorously? 

Reviewer #1: Yes

Reviewer #2: Yes

4. Have the authors made all data underlying the findings in their manuscript fully available?

Reviewer #1: Yes

Reviewer #2: Yes

5. Is the manuscript presented in an intelligible fashion and written in standard English?

Reviewer #1: Yes

Reviewer #2: Yes

6. Review Comments to the Author

Reviewer #1: (No Response)

Reviewer #2: The authors have addressed all the comments and the paper reads very well. I am fully satisfied with the revised manuscript.

7. PLOS authors have the option to publish the peer review history of their article (what does this mean?). If published, this will include your full peer review and any attached files.

Reviewer #1: Yes: Dr. Ahbab Mohammad Fazle Rabbi

Reviewer #2: Yes: Kannan Navaneetham, Professor, Department of Population Studies, University of Botswana, Botswana

---

## [Editor Report · Acceptance letter]

12 Nov 2019

PONE-D-19-14424R1 

Is half the world’s population really below ‘replacement-rate’? 

Dear Dr. Gietel-Basten:

I am pleased to inform you that your manuscript has been deemed suitable for publication in PLOS ONE. Congratulations! Your manuscript is now with our production department. 

With kind regards,

on behalf of

Professor Hafiz T.A. Khan 

Academic Editor

PLOS ONE